# Food-based indices for the assessment of nutritive value and environmental impact of meals and diets: A systematic review protocol

**Eva-Leanne Thomas**[1]*, **David Livingstone**[1], **Anne P. Nugent**[1,2], **Jayne V. Woodside**[1,3], **Paul Brereton**[1]

**1** Institute for Global Food Security, Queen's University Belfast, Belfast, United Kingdom, **2** School of Agriculture and Food Science, UCD Institute of Food and Health, University College Dublin, Ireland, **3** School of Medicine, Dentistry and Biomedical Sciences, Queen's University Belfast, Belfast, United Kingdom

* Ethomas08@qub.ac.uk

**Data Availability Statement:** No datasets were generated or analysed during the current study. All

## Abstract

Current food production and consumption practices are impacting both human and planetary health. Though these challenges are multifaceted, shifting to healthy dietary choices from sustainable food systems is one solution. Food-based labelling is a common public-health strategy aimed at influencing consumption practices, primarily displaying front-of-pack nutrition labelling to encourage healthier choices. Due to the dual impact of food on human and planetary health there is a growing need to additionally include environmental impact information. However, this potentially conflicting information could overwhelm consumers, necessitating a simplified approach that combines both nutritive and environmental values. Previous work has identified existing models, termed sustainable food profiling models, for labelling of individual food products based on their environmental impacts and nutritive value. Foods are rarely eaten in isolation and are often consumed as composite meals, which aggregate to diets. Therefore, it is important to identify indices that exist to assess or rank meals and diets according to their nutritional and environmental impacts. Following the Preferred Reporting Items for Systematic Review and Meta-Analysis Protocols (PRISMA-P) 2015 guidelines, our systematic review protocol was registered with the International Prospective Register of Systematic Reviews (PROSPERO) on 01 May 2024 (PROSPERO registration ID = CRD42024537149). In the present protocol the methodology to identify and review existing food-based indices for the assessment of nutritive value and environmental impact of meals and diets and their intended public health purpose is described. Our primary research questions are: (i) what food-based indices assessing nutritive value and environmental impact exist for classification or ranking of meals or diets? and (ii) what are the methods used to create, and the key characteristics of, these indices?. The results are expected to increase understanding of, and highlight the variation in, the creation of combined measures for the assessment of nutritive value and environmental impact for meals and diets. Additionally, findings can be used to inform researchers, business, and policy actors on future approaches for labelling foods, meals, or diets in a way that supports healthy and sustainable meal choices and diets.

relevant data from this study will be made available upon study completion.

**Funding:** This systematic review is funded by a Postgraduate Research Studentship from the Department for the Economy, Northern Ireland. Funders had no role in the development of the protocol.

**Competing interests:** The authors have declared that no competing interests exist.

## Introduction

Human and planetary health are negatively impacted by current food production and consumption practices. Food production and agriculture contributes to over a quarter of the world's anthropogenic greenhouse gas emissions, occupies around half the world's habitable land, and uses 70% of valuable global freshwater withdrawals [1–4]. Simultaneously, current dietary intakes across developed and developing nations has caused the double burden of malnutrition, defined as the simultaneous manifestation of both undernutrition and overweight and obesity [5]. One of the five outline strategies by the World Health Organization to act across the food system and work towards tackling these issues is to shift to sustainable consumption patterns by promoting and creating demand for healthy and sustainable diets [6]. Sustainability, in relation to food production and consumption, incorporates an increasing number of dimensions [7]. Sustainable diets are defined by the FAO as "*those diets with low environmental impacts which contribute to food and nutrition security and to healthy life for present and future generations. Sustainable diets are protective and respectful of biodiversity and ecosystems, culturally acceptable, accessible, economically fair and affordable; nutritionally adequate, safe and healthy; while optimizing natural and human resources.*" [8]. Though frameworks outlining global food system transformation for planetary and human health exist, one of the barriers to shifting towards sustainable diets is limited consumer knowledge and awareness of what constitutes such diets [9, 10].

Food nutrition labelling is a commonly used public health tool that aims to inform consumer choice and shift consumption patterns [11]. Due to the growing awareness of the need for healthy diets from sustainable food systems, there is now an increasing interest across the food system to include the environmental impacts of food on front-of-pack (FOP) labelling, known as eco-labelling [12]. However, an unintended consequence of eco-labelling is the health halo effect, where consumers have been found to misperceive foods with eco-labelling as healthier, though low environmental impacts and healthfulness are not mutually exclusive [13]. Presenting both the nutritive value and environmental impacts of foods on FOP labelling could mitigate this health halo effect and is a potential way to overcome the barrier of low consumer knowledge of sustainability of foods.

Labelling initiatives combining nutritive value and environmental impacts of foods are now appearing within the literature. A previous systematic review identified ten food labelling initiatives that scored individual foods according to at least two environmental impacts, of which six additionally scored individual foods based on nutritional quality, which authors termed sustainable food profiling models [14]. The decision to focus on nutritive value and environmental impacts as opposed to the all-encompassing definition of sustainability comes from the lack of consensus on wider dimension definitions alongside immature understanding and supporting standards for such dimensions to be used in a pragmatic index, which could further confuse the consumer. For the purposes of this protocol these labelling initiatives will be referred to as nutritive and environmental combined indices.

Moreover, foods are not consumed on an ingredient level, they are predominantly consumed as composite meals, which aggregate to reflect an individual's diet. Therefore, it is important to identifying nutritive and environmental combined indices designed for the assessment of meals and diets, beyond individual foods. Bunge and colleagues highlighted how research filling this gap could act as a public health tool potentially increasing informed consumer choice in settings such as restaurants or to underpin public health policy, such as menu reformulation or marketing restrictions [14].

These underline the motivation for the current systematic review that aims to provide insights into the food-based indices that exist that assess both nutritive value and

environmental impacts of meals and diets, to highlight the methodological variety in the design of these indices and outline their intended public health purposes.

The key PICO (Population, Intervention, Comparison and Outcomes) framework [15] has been adapted to the following systematic review. There is no restriction on population as nutritive and environmental combined indices for the assessment or ranking of meals or diets should be applicable and suitable for all population groups. Nutritive value and environmental impact indices for assessment or ranking of meals and diets are considered the intervention with no comparator. the outcomes considered are the number of combined indices including a nutrition score dimension and an environmental impact dimension for the assessment or ranking of meals or diets, methodology of combination, key characteristics, and public health purpose.

The present study protocol is described to understand what food-based indices assessing nutritive value and environmental impacts exist for classification or ranking of meals or diets? And what are the methods used to create, and the key characteristics of, these indices?

Nutrient profiling, defined as "the science of classifying or ranking foods according to their nutritional composition for reasons related to preventing disease and promoting health" [16], is used to define the nutritive value of foods. There is no definition for environmental impacts profiling models, however, Environmental Impact Assessment (EIA) can be used and is broadly defined by the European Commission as "a study of the effects of a proposed project, plan or program on the environment", [17]. More specifically, environmental impacts in relation to this systematic review can be described as a metric (e.g. numeric score or letter) quantifying the effect of meals and diets on the natural environment.

The secondary research question for this systematic review is to explore what the intended public health purpose of these indices are.

## Materials and methods

This systematic review protocol follows the PRISMA-P (Preferred reporting items for systematic review and meta-analysis protocols) checklist for recommended items to address in a systematic review protocol [18, 19]. The 17 items that are considered necessary to report a systematic review protocol are formed in a checklist (S1 File) [18].

### Bibliographic databases and database management

A structured search was conducted in six bibliographic databases: MEDLINE, EMBASE, Web of Science (science and social science citation index), Scopus, CAB abstracts and Food Science and Technology Abstracts, in April 2024. These databases were identified as the most appropriate and relevant to the research field. The database searches were done under the existing subscriptions of the organisations of the authors.

### Search string scoping

Relevant papers were identified from the reference lists of previous systematic and non-systematic reviews [14, 20]. Keywords from these relevant papers were identified and grouped into themes based on the PICO elements of this systematic review (S1 Appendix in S2 File). The most frequently used keywords were used to adapt and update a previously reported word string [14]. To optimize the search string a scoping exercise was carried out in Web of Science where different key words and narrower or wider search strings were tested. A list of the most relevant papers identified in each search was created and the retrieved results from each trialed word string were compared to this list to identify the most appropriate search strategy. Key papers identified as work of Rs et al. (2015) [21], Seconda et al. (2019) [22] and van Dooren

et al. (2014) [23]. This enabled a balance between specificity (reducing the number of irrelevant studies) and sensitivity (ensuring the maximum number of relevant studies were identified) to be struck. The search string that identified the key papers was used to search the selected bibliographic databases.

## Search string formulation

A search string using the Boolean operators OR, AND, and, AND NOT was used to conduct a search in all six bibliographic databases to retrieve relevant literature that combines terms for the following components: nutritive value, environmental impact, model, and meal / diet, and removes terms related to animals (Table 1). The search string shown in Table 1 was adapted to the requirements of the abovementioned databases, but the same terms and search fields were used (title, abstract and keywords). The search string details for each database can be found in the S2 Appendix in S2 File.

## Article screening

**Screening strategy.**  EndNote version 21.4 will be used to catalogue retrieved articles and will be used to remove duplicates where more than one bibliographic database has returned the same article. Article data will be exported from EndNote, a reference management software package, and imported into Ryyan, a web-based systematic review software, for a two-staged screening strategy by two independent, blinded authors.

The first stage will assess the relevance of each article based on the title and abstract. Articles that clearly meet the exclusion criteria will be excluded and articles that seemingly meet the inclusion criteria will be included for stage two screening. This will involve assessment of article relevance through a review of their full text. Articles that meet any of the exclusion criteria will be excluded, with the exclusion reasons during this stage being noted for each article. If additional information is mandatory to determine eligibility, up to three attempts to contact the original study authors will be made. If no response is received, the paper will be excluded. Once all full text articles have been screened blinding will be turned-off and conflicts will be discussed between the two reviewers. Where there is uncertainty, the article will be marked for review by a third author.

Ryyan provides the number of conflicts between authors, from this a decimal (amount of agreements / number of papers screened) can be calculated. This value can be used as the Kappa value and a Cohen's kappa can be interpreted [24].

**Eligibility criteria.**  The PICO framework was used to develop the eligibility criteria to ensure only relevant articles are included in the final analysis. The following inclusion and exclusion criteria (Table 2) will be used to screen retrieved studies:

• Study type:

The criteria will include relevant observational studies, consumer experiment studies, modelling and optimization studies as well as protocol and methodological studies. To ensure consistency and accessibility, only full-text peer-reviewed articles published, and not withdrawn by authors, in the last 15-years will be included (2009–2024).

• Participants:

No specific group of participants or populations will be included or excluded as meals and diets that are nutritionally adequate that have a low environmental impacts should not be restricted. Policies, initiatives, and concepts combining nutritive value and environmental

**Table 1. Search string for bibliographic databases (four components will be combined with the Boolean operator AND before combination with the Boolean operator 'NOT' for the fifth component).**

| Nutritive Value | AND | Environmental Impact | AND | Model | AND | Meal / Diet | AND NOT | Animals |
|---|---|---|---|---|---|---|---|---|
| nutri* NEAR/2 (quality or footprint or food* or profiling or density or score or index or adequacy)) or (food* NEAR/2 (label* or suppl* or consumption or choice or environment or pattern*)) or (eating NEAR/2 (indicator or score* or impact*)) or (feeding behavio $r* NEAR/2 (indicator or score* or impact*)) or (public health NEAR/2 (indicator or score* or impact*)) or (health* NEAR/2 (indicator or score* or impact*)) | | (sustainab* NEAR/2 (impact or assess* or evaluat* or indicator? or health*)) or (environment* NEAR/2 (impact or assess* or evaluat* or indicator? or health*)) or (climate* NEAR/2 (impact or assess* or evaluat* or indicator? or health*)) or (land* NEAR/2 (clear* or "use*" or usage)) or (soil NEAR/2 (clear* or "use*" or usage)) or (water NEAR/2 ("use*" or usage or foot*)) or (life cycle NEAR/2 (evaluation or assessment)) or eutrophication or (carbon NEAR/2 (foot* or ecosystem)) or (fossil NEAR/2 fuels) | | (model* or algorithm* or metric* or scor* or rank* or framework* or index or indices or tool* or inventor* or validation or "multi-criteria analys*" or guideline* or impact* or optim*) | | (meal* or menu* or recipe* or diet*) | | (animals or animal or mice or mus or mouse or murine or woodmouse or rats or rat or murinae or muridae or cottonrat or cottonrats or hamster or hamsters or cricetinae or rodentia or rodent or rodents or pigs or pig or swine or swines or piglets or piglet or boar or boars or sus scrofa or ferrets or ferret or polecat or polecats or mustela putorius or guinea pigs or guinea pig or cavia or callithrix or marmoset or marmosets or cebuella or hapale or octodon or chinchilla or chinchillas or gerbillinae or gerbil or gerbils or jird or jirds or merione or meriones or rabbits or rabbit or hares or hare or diptera or flies or fly or dipteral or drosphila or drosophilidae or cats or cat or carus or felis or nematoda or nematode or nematoda or nematode or nematodes or sipunculida or dogs or dog or canine or canines or canis or sheep or sheeps or mouflon or mouflons or ovis or goats or goat or capra or capras or rupicapra or chamois or haplorhini or monkey or monkeys or anthropoidea or anthropoids or saguinus or tamarin or tamarins or leontopithecus or hominidae or ape or apes or pan or paniscus or pan paniscus or bonobo or bonobos or troglodytes or pan troglodytes or gibbon or gibbons or siamang or siamangs or nomascus or symphalangus or chimpanzee or chimpanzees or prosimians or bush baby or prosimian or bush babies or galagos or galago or pongidae or gorilla or gorillas or pongo or pygmaeus or pongo pygmaeus or orangutans or pygmaeus or lemur or lemurs or lemuridae or horse or horses or pongo or equus or cow or calf or bull or chicken or chickens or gallus or quail or bird or birds or quails or poultry or poultries or fowl or fowls or reptile or reptilia or reptiles or snakes or snake or lizard or lizards or alligator or alligators or crocodile or crocodiles or turtle or turtles or amphibian or amphibians or amphibia or frog or frogs or bombina or salientia or toad or toads or epidalea calamita or salamander or salamanders or eel or eels or fish or fishes or pisces or catfish or catfishes or siluriformes or arius or heteropneustes or sheatfish or perch or perches or percidae or perca or trout or trouts or char or chars or salvelinus or fathead minnow or minnow or cyprinidae or carps or carp or zebrafish or zebrafishes or goldfish or goldfishes or guppy or guppies or chub or chubs or tinca or barbels or barbus or pimephales or promelas or poecilia reticulata or mullet or mullets or seahorse or seahorses or mugil curema or atlantic cod or shark or sharks or catshark or anguilla or salmonid or salmonids or whitefish or whitefishes or salmon or salmons or sole or solea or sea lamprey or lamprey or lampreys or pumpkinseed or sunfish or sunfishes or tilapia or tilapias or turbot or turbots or flatfish or flatfishes or sciuridae or squirrel or squirrels or chipmunk or chipmunks or suslik or susliks or vole or voles or lemming or lemmings or muskrat or muskrats or lemmus or otter or otters or marten or martens or martes or weasel or badger or badgers or ermine or mink or minks or sable or sables or gulo or gulos or wolverine or wolverines or minks or mustela or llama or llamas or alpaca or alpacas or camelid or camelids or guanaco or guanacos or chiroptera or chiropteras or bat or bats or fox or foxes or iguana or iguanas or xenopus laevis or parakeet or parakeets or parrot or parrots or donkey or donkeys or mule or mules or zebra or zebras or shrew or shrews or bison or bisons or buffalo or buffaloes or deer or deers or bear or bears or panda or pandas or wild hog or wild boar or fitchew or fitch or beaver or beav or jerboa or jerboas or capybara or capybaras) |

**Table 2. Inclusion and exclusion criteria.**

| | Inclusion Criteria | Exclusion Criteria |
|---|---|---|
| Study Design | Published 2009–2024<br>Peer-reviewed articles<br>Quantitative empirical studies<br>Full-text articles | Non-peer reviewed articles<br>Review works or book chapters<br>Qualitative studies<br>Retracted articles<br>Abstract only articles |
| Participants | No specific group of participants or populations will be included | No specific group of participants or populations will be excluded |
| Intervention | Indices assessing nutritive value and environmental impacts of meals or diet<br>Indices for the ranking, or classification of a meal or diet | No index specified<br>Ranking or classification of meals and diets according to exclusively nutritional composition or environmental impacts<br>Indices appropriate for the use of individual food items or food system only<br>Index already included |
| Outcomes | Indices including qualitative nutritive and environmental scores<br>Index methodology for the combination, assessment, or ranking of the two dimensions outlined<br>Describes index public health purpose | Index methodology for the combination, assessment, or ranking of the two dimensions not described<br>Indices using quantitative analysis for nutritive or environmental scoring<br>Public health purpose is not distinguishable<br>Index public health purpose not referred to |

impacts for the assessment or ranking of meals or diets should be applicable and suitable for all population groups.

- Intervention:

The criteria will review indices (including models, tools, or frameworks) that include the nutritive value and at least one environmental impacts appropriate for the assessment or ranking of meals or diets. Therefore, studies describing the nutritive or environmental impacts in relation to meals or diets without presenting this as an index will be excluded. Studies will also be excluded if they assess nutritive value or environmental impacts solely, refer to an index that has already been included in the review, for example a validation or follow-up study, or detail an index that does not have scope to be used for the assessment or ranking of meals or diets, for example are designed specifically for the assessment of individual food items or food systems.

- Outcomes:

The number of combined indices including a nutrition score dimension and an environmental impacts dimension for the assessment or ranking of meals or diets as well as the methodology of dimension combination and key characteristics of the index are the main outcomes of this review.

Therefore, included studies need to specify nutritional quality and at least one environmental impacts of a meal or diet quantitatively. Indices that use qualitative measures (e.g., scoring based on dietary behaviours) to define nutritional quality or environmental impacts of meals or diets will be excluded. Studies will also be excluded if there is no clear methodology outlining the combination, assessment, or ranking of the two dimensions.

A secondary outcome for this review is to define the public health purpose of the index. Hence, studies that do not refer to the intended public health purpose of the index will be excluded.

## Study records

**Study validity assessment.** This review will narratively synthesise the number of and existing approaches to the creation of indices that simultaneously assess nutritive value and environmental impacts of meals and diets. All studies that meet the inclusion criteria will be

included in the synthesis regardless of their risk of bias. The methodological quality of the studies will be assessed using an amended criteria previously defined by Bunge et al. (2021), known as the model replicability assessment [14], re-termed the index replicability assessment and included in the data extraction spreadsheet. Replicability will be assessed on a scale from 1 (replicable), where equations, system boundaries, final scoring and cut-offs are included and data sets are referenced and fully available, to 7 (not replicable), where the methodology is not described, no equations, system boundaries or final scoring is included, and datasets are not referenced.

Two authors will be involved in the quality assessment. Any discrepancies will be discussed and resolved via consensus. If consensus cannot be reached, a senior reviewer will review the eligible paper and make a final decision.

**Data extraction (data categorisation).** Information on studies included in this review will be extracted and collected in a piloted data extraction Excel spreadsheet, which will be accessible to all study authors. The following data, if available, will be extracted from each eligible paper based on the PICO framework for this review:

1. Index name

2. Year indices was launched / published

3. Country of publication

4. Type and name of institution

5. Geographical scope

6. Aim

7. Target population

8. Application (e.g., (A) meal, (B) diet, or (C) scope for meals/diets

9. Index classification (e.g. (A) composite / combined single index or (B) multiple-score index)

10. Dimensions (e.g. (A) nutritive value and environmental impacts or (B) nutritive value, environmental impacts, and other factor (e.g., economic or socio-cultural consideration)

11. Name of nutritional score and equation

12. Name of environmental score and equation

13. List of nutrient values

14. List of environmental impact indicators

15. System boundaries

16. Reference amount

17. Approach to combining nutritional and environmental components (e.g. algorithm or calculation and weighting)

18. Cut off value for healthy meal/diet

19. Cut-off value for environmentally friendly meal/diet

20. Cut off value for healthy and environmentally friendly meal/diet

21. Public health purpose

**Table 3. Data categorisation of data extraction points.**

| Categories | Items |
|---|---|
| General Information | 1–6, 22 |
| Participants | 7 |
| Intervention | 8–10 |
| Outcomes | 11–21 |

22. Database(s)

Table 3 shows how the data extraction points will be categorised.

To ensure repeatability the primary reviewer will extract data from all articles and a second reviewer will extract data from every 5[th] article and data extraction will be cross-checked. Any discrepancies will be discussed and resolved via consensus. If consensus cannot be reached, a senior reviewer will review the eligible paper and make a final decision. All extracted data will be included in the systematic review manuscript.

## Data synthesis

Synthesising extracted data is a key element of a systematic review. Statistical analysis will not be carried out due to the qualitative nature of data that will be collected and therefore conducting a narrative synthesis will allow authors to understand how the interventions work, why and for who, enabling exploration of relationships in the data and allowing for strength of evidence assessment in order to draw evidence based conclusions [25]. Counts will be used to report the number of identified indices and a heterogenous table will be used to report the general information. Summary and matrix tables will be used to present extracted data relating to the intervention and outcomes with supporting narrative synthesis to enable comparisons between the indices included in the review, and to answer the primary and secondary research questions.

**Risk of publication bias.** The authors have no conflicts of interest. Two reviewers, with an additional third reviewer where necessary, will participate in study screening and data extraction and minimum threshold of substantial agreement according to Cohen's kappa [24] will be used to reduce selection bias.

## Conclusion

Eating habits are a key modifiable factor influencing personal and planetary health. By identifying the existing indices and assessing the methodology and characteristics alongside their intended public health purpose the results of the systematic review can be used to inform researchers, business, and policy actors in the future development of an index for the purpose of labelling foods, meals or diets in a way that reflects both human and planetary health. For example the adaptation of public health policies or initiatives, such as highlighting the need for meal reformulation or marketing restrictions to support healthy and sustainable meal and diet choices.

Main limitations of this study include the varied methodology used to create nutritive and environmental combined indices as well as the diverse range of key information included in these indices making direct comparison difficult. However, carrying out this systematic review could highlight where the variance in how these indices are created lies and demonstrate the need for standardisation of such indices.

Additionally, there is no exclusion of papers based on language, which may lead to the need for papers to be translated to English. Translation of technical language can be difficult and

can lead to the translated text not conveying the intended meaning of the original text or make understanding the original text difficult and therefore could be another limitation in our study. The reasoning behind this is to ensure a thorough search of the literature making sure to capture all combined nutritive and environmental indices regardless of language.

## Supporting information

**S1 File. PRISMA-P checklist.**
(DOCX)

**S2 File. Search string development and details for each database.**
(DOCX)

## Acknowledgments

We would like to thank Patrick Elliot, Subject Librarian–Biological Science at Queen's University who assisted with word string formulation and database selection.

## Author Contributions

**Conceptualization:** Eva-Leanne Thomas, Anne P. Nugent, Jayne V. Woodside, Paul Brereton.

**Methodology:** Eva-Leanne Thomas, David Livingstone.

**Writing – original draft:** Eva-Leanne Thomas.

**Writing – review & editing:** David Livingstone, Anne P. Nugent, Jayne V. Woodside, Paul Brereton.

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
