## [Decision Letter · Decision Letter 0]

22 Nov 2024

PONE-D-24-36565Food-based indices for the assessment of nutritive value and environmental impact of meals and diets: A systematic review protocolPLOS ONE

Dear Dr. Thomas,

Thank you for submitting your manuscript to PLOS ONE. After careful consideration, we feel that it has merit but does not fully meet PLOS ONE’s publication criteria as it currently stands. Therefore, we invite you to submit a revised version of the manuscript that addresses the points raised during the review process.

We look forward to receiving your revised manuscript.

Kind regards,

Sayyed Mohammad Hadi Alavi

Academic Editor

PLOS ONE

2. As required by our policy on Data Availability, please ensure your manuscript or supplementary information includes the following:

Additional Editor Comments:

PONE-D-24-36565

Reviewed by: Hadi Alavi, AE, PLoS ONE

Authors have designed a study protocol to investigate Food-based indices for the assessment of nutritive values and environmental impacts of meals and diets. Results provide very valuable information to public and governmental sectors to support healthy and sustainable meal choices and diets.  The MS was well designed, however it needs revision before being accepted for publication.

Abstract should be shorten. It is too long for the necessity and strength of the study. Please fix it in limited number of words. I believe it to be possible with max. 350 words.

L28: to express

L28: and should not be Italic.

L28: values

L37: impacts

L39: comma after guidelines

L42: in the present study as a substitute for here

L42: methodology to review food-based indices determined for ….

L49-53: Outcomes could be shorten.

L68: comma after land

L76: I think the best is to use environmental impacts through the MS. It is better than environmental impact. >>> L87, L89, L91, L112, L205 and more

L98: Similar to former comment, I think nutritive values  could be considered as a substitute for nutritive value  through the MS. >>> L112, L205

L111: These as a substitute for this (Many reason are aforementioned to stress strength of the present study).

L116 – 125: Please re-structure. Use of bullets is not common.

L126 – 130: Please re-structure. You may use: The present study was conducted to understand What food-based indices ….. and what are the methods used

L138 – 139: Pleas re-structure similar to previous comment

L144: formed in a checklist (Supplementary material – File #1) [18].  

L150 – 151: Please omit “Search results will be downloaded and catalogued in EndNote version 21.3.”

L155: Supplementary material (File #2, appendix S1). Please use this formulation for citation of the supplementary material in the text. >>> L171

L157-158: please omit (science and social science index)  

L161: Rӧӧs et al. [21], Seconda et al. [22] and van Dooren et al. [23].  

L172: Search string for bibliographic databases  

L180, L190, L240: reviewers? It may better to use “authors”. I am wondering reviewers are incorporated into study design

L195: will be used

L198: It may be better to use This criteria as a substitute for this review  

L208: This criteria will review …

L282: Data synthesis : Please describe this section in details. What will be done with data including statistical tests that may be useful to answer the questions (hypotheses)

L294 – 300: Authors may stress the impacts of nutritional values on health. For instance there are frequent studies highlighting the impacts of high-fat-diets on prevalence of diabetes, cognitive disorders and ….

L301 – 304: This is very important limitation, please discuss in detail. It might be used as a criteria to determine the eligibility of a study to be incorporated into Meta-analysis. For instance, if sample size for determination of nutritional values are well enough, if grouping of foods for determination f nutritional values was rational, etc

Reviewers' comments:

Reviewer's Responses to Questions

**Comments to the Author**

1. Does the manuscript provide a valid rationale for the proposed study, with clearly identified and justified research questions?

Reviewer #1: Yes

2. Is the protocol technically sound and planned in a manner that will lead to a meaningful outcome and allow testing the stated hypotheses?

Reviewer #1: Yes

3. Is the methodology feasible and described in sufficient detail to allow the work to be replicable?

Reviewer #1: Yes

4. Have the authors described where all data underlying the findings will be made available when the study is complete?

Reviewer #1: No

5. Is the manuscript presented in an intelligible fashion and written in standard English?

Reviewer #1: Yes

6. Review Comments to the Author

You may also provide optional suggestions and comments to authors that they might find helpful in planning their study.

Reviewer #1: The manuscript presents clearly identified and justified research questions, and the methods provide sufficient detail to allow for reproduction.

Supporting information on PRISMA-P check list and search strings details for each database are accessible.

The protocol can be accepted for publication.

7. PLOS authors have the option to publish the peer review history of their article (what does this mean?). If published, this will include your full peer review and any attached files.

Reviewer #1: No

---

## [Author Response · Author response to Decision Letter 0]

2 Dec 2024

Dear Editor,

Thank you for coordinating the review process and providing constructive feedback. We have carefully addressed the comments from both the reviewers and yourself. The details of our responses can be found in the attached Response to Reviewers_ELT_PO rebuttal letter, and all revisions have been tracked in the accompanying Revised Manuscript with Track Changes_PO_ELThomas.

We appreciate the opportunity to improve our manuscript and we are pleased to resubmit it for your consideration.

Kind regards,

Eva-Leanne Thomas

---

## [Editor Report · Decision Letter 1]

3 Dec 2024

Food-based indices for the assessment of nutritive value and environmental impact of meals and diets: A systematic review protocol

PONE-D-24-36565R1

Dear Dr. Thomas,

We’re pleased to inform you that your manuscript has been judged scientifically suitable for publication and will be formally accepted for publication once it meets all outstanding technical requirements.

Kind regards,

Sayyed Mohammad Hadi Alavi

Academic Editor

PLOS ONE
---

## [Editor Report · Acceptance letter]

6 Dec 2024

PONE-D-24-36565R1 

PLOS ONE

Dear Dr. Thomas, 

I'm pleased to inform you that your manuscript has been deemed suitable for publication in PLOS ONE. Congratulations! Your manuscript is now being handed over to our production team.

Kind regards, 

on behalf of

Dr. Sayyed Mohammad Hadi Alavi 

Academic Editor

PLOS ONE